# SDN-Based Congestion Control and Bandwidth Allocation Scheme in 5G Networks

**DOI:** 10.3390/s24030749

**Published:** 2024-01-24

**Authors:** Dong Yang, Wei-Tek Tsai

**Affiliations:** 1School of Computer Science and Engineering, Beihang University, Beijing 100191, China; tsai@tiandetech.com; 2Fuzhou FuYao Institute for Advanced Study, Fuzhou 350300, China

**Keywords:** congestion control, SDN, routing feedback, in-network control, 5G cellular network

## Abstract

5G cellular networks are already more than six times faster than 4G networks, and their packet loss rate, especially in the Internet of Vehicles (IoV), can reach 0.5% in many cases, such as when there is high-speed movement or obstacles nearby. In such high bandwidth and high packet loss network environments, traditional congestion control algorithms, such as CUBIC and bottleneck bandwidth and round-trip propagation time (BBR), have been unable to balance flow fairness and high performance, and their flow rate often takes a long time to converge. We propose a congestion control algorithm based on bottleneck routing feedback using an in-network control mode called bottleneck routing feedback (BRF). We use SDN technology (OpenFlow protocol) to collect network bandwidth information, and BRF controls the data transmission rate of the sender. By adding the bandwidth information of the bottleneck in the option field in the ACK packet, considering the flow fairness and the flow convergence rate, a bandwidth allocation scheme compatible with multiple congestion control algorithms is proposed to ensure the fairness of all flows and make them converge faster. The performance of BRF is evaluated via Mininet. The experimental results show that BRF provides higher bandwidth utilization, faster convergence rate, and fairer bandwidth allocation than existing congestion control algorithms in 5G cellular networks.

## 1. Introduction

In the past decade, the data volume of the cellular network has continued to increase, from 3G networks to 4G networks and then to 5G networks. According to statistics, the average speed of the existing 4G network is 25 Mbps [1], the average speed of the 5G network is 150 Mbps [2], and the peak speed of the 5G network can even reach 1 Gbps. 5G cellular networks also have far more base stations than 4G networks, resulting in increasingly complex network topologies. The bandwidth of the 5G core network is also much larger than that of the 4G network, which is caused by the increase in 5G base station bandwidth and the number of 5G base stations. As the terminals of the 5G network are mainly mobile objects such as mobile phones and vehicles, the network packet loss rate will be very high in the case of base station switching or scenarios with obstacles nearby. In such a high-bandwidth and high-jitter network environment, it is extremely easy for a buffer queue jam to occur on the bottleneck route, which causes a reduction in the network bandwidth utilization rate and an increase in delay. The existing congestion control (CC) algorithms can only solve part of the problem and have many shortcomings.

CC algorithms are the key to preventing network congestion. They are particularly important in 5G networks. CUBIC [3] is a loss-based CC algorithm, and it is the default CC algorithm in Linux servers. CUBIC has good performance on wired networks. However, once packet loss occurs, the data transmission speed of the server slows, and CUBIC will cause insufficient bandwidth utilization in wireless networks where packet loss occurs frequently [4]. BBR [5] is a new CC algorithm that was proposed by Google in 2016. It builds an explicit model of the network using the estimated RTT (round-trip time) and the estimated available bottleneck bandwidth to prevent congestion. BBR overcomes the shortcomings of the CUBIC algorithm, but it causes unfairness and many packet losses when competing with loss-based CC algorithms [6]. The above algorithms are all end-to-end CC algorithms. The ECN [7] and XCP [8] protocols are algorithms used in in-network control modes. The in-network control mode has finer control power than the end-to-end mode and can directly obtain the queue status of the router. However, the ECN only takes effect when the network is congested. The CE flag notifies the sender to reduce the transmission rate. When the network is idle, the ECN does not instruct the sender to increase the transmission rate. The source end of the ECN protocol still responds to the phenomenon of congestion in the network rather than to the degree of congestion [9], and the ECN cannot accurately control the sending rate. The stability of the XCP protocol depends on the accuracy of estimating the number of flows currently in the network, but it is difficult to accurately estimate the number of flows. Once the estimated value is too small, it is easy for burst traffic to occur on the network, and if the estimated value is too large, a low link utilization rate can easily occur [10]. The above protocols are the latest in the field of congestion control, but none of them can meet the requirements of the 5G network, and they cannot achieve a balance in bandwidth utilization, fairness, and convergence speed.

A software-defined network (SDN) is a network architecture whose core idea is to separate network control and forwarding. In a software-defined network, a network engineer or administrator can adjust traffic from a central console without having to touch individual switches in the network, and a centralized SDN controller directs the switch to provide network services wherever needed, regardless of the specific connections between servers and devices. At present, there are discussions on the CC mechanism in SDN-based networks [11,12], but its CC mechanism is still based on an end-to-end mode, which cannot exploit the centralized control mode of an SDN.

In this paper, we propose a congestion control algorithm BRF using an in-network control mode based on bottleneck routing feedback and SDN. The OpenFlow controller is used to collect network card information [13], and OpenSwitch is deployed at the bottleneck using bridge mode, which does not change the topology of the existing network. The Linux kernel with the BRF congestion control algorithm runs on the switch and writes the feedback information of each flow by using two extra bytes of the option field in ACK. The feedback stores the bandwidth allocated to the flow by the bottleneck route. The bottleneck route recalculates the ACK checksum and forwards the ACK. After receiving the ACK, the data sender reads the feedback information and calculates the pacing rate of the flow to control the data transmission rate. BRF uses a centralized control mode to uniformly allocate bandwidth for each flow at the bottleneck. This ensures that network congestion does not occur and that the link bandwidth is fully utilized. Additionally, we propose a hierarchical bandwidth allocation scheme based on the in-network control mode to ensure fair bandwidth allocation among flows. This involves stratifying all flows according to their maximum data transmission speed and dynamically adjusting each flow’s level to prioritize small-bandwidth flow bandwidth requirements. For newly added flows, a bandwidth value is initially allocated based on the network quintuple. Subsequently, a suitable bandwidth is assigned to the flow according to the test results, ensuring an average data flow convergence speed of 1 RTT. The allocation of bandwidth adopts a centralized control mode based on SDN to ensure fairness among all flows.

In summary, our innovative work on congestion control algorithms in 5G networks in this paper mainly includes two parts: a congestion control algorithm combining SDN and an in-network control mode and a hierarchical bandwidth allocation scheme. The first part aims to improve bandwidth utilization and reduce queueing delays. The second part aims to ensure the fairness of each flow and improve the convergence speed. The main contributions of our work can be further summarized as follows:To improve the bandwidth utilization of 5G networks, we propose a congestion control algorithm that combines SDN and an in-network control mode. The data transmission speed can be precisely controlled through the router feedback mode to avoid the disadvantage of the end-to-end mode, which is that the adjustment of the sending speed on the sender always lags behind the actual situation. Compared with ECN and XCP protocols, BRF directly feeds back the allocated bandwidth and can more accurately adjust the data transmission speed. The BRF can precisely control the total bandwidth of each flow so as not to exceed the bottleneck bandwidth to avoid router packet loss and queueing.To ensure the fairness of each flow and the speed of flow convergence, we propose a hierarchical bandwidth allocation scheme based on the in-network control mode. First, the initial bandwidth of each flow is allocated according to the network quintuple. Then, after a measurement, the flow is stratified according to the measurement results. Each layer has its own data-sending queue, and queues in different layers are assigned different priorities. Mouse data flows are allocated to the higher priority but less bandwidth-consuming layer, while elephant data flows are allocated to the lower priority but more bandwidth-consuming layer. The bandwidth of data flows within the same layer is evenly allocated and adjusted adaptively.We conduct experiments with Mininet. We compared BRF with existing CC algorithms, including CUBIC and BBR, in terms of bandwidth utilization, queue delay, routing packet loss rate, fairness, and bandwidth convergence speed. The experimental results show that BRF is superior to the existing CC algorithm in all five aspects.

The remainder of this paper is structured as follows. Section 2 reviews related work on the CC algorithm and the evaluation index of the CC algorithm. In Section 3, we present the design of the BRF algorithm and the bandwidth allocation scheme. The performance analysis is given in Section 4, and Section 5 gives the concluding remarks of this work and discusses future work.

## 2. Related Work

CC algorithms are one of the key factors in determining network performance. After decades of development, existing CC algorithms are mainly divided into two categories, one based on the end-to-end mode [14] and the other based on the in-network control mode [15]. The CC algorithm based on the end-to-end mode treats the network as a black-box system. A series of adaptive algorithms are used to adjust the data transmission rate and window value of the sender. The CC algorithm based on the in-network control mode treats the network as a white-box system, which can obtain information such as the congestion situation in the network. The following is a brief introduction to both types of algorithms.

### 2.1. CC Algorithm Based on the End-to-End Mode

CC algorithms based on the end-to-end mode, which refers to direct control of the data transmission speed by the data transmitter, are mainly divided into three categories: loss-based CC algorithms, delay-based CC algorithms, and hybrid CC algorithms.

#### 2.1.1. Loss-Based CC Algorithms

Loss-based CC algorithms adjust the data transmission rate and window value based on the packet loss signal. The Reno algorithm is the earliest loss-based CC algorithm, which uses a slow start, congestion avoidance, fast retransmission, and fast recovery mechanism, which is the basis of many existing algorithms. NewReno is an improved version based on Reno, mainly improving the fast recovery algorithm; in the NewReno algorithm, only when all the lost packets are retransmitted and received does the algorithm acknowledge successful transfer. In NewReno, a recovery response judgment function is added, which enables TCP terminals to distinguish between multiple packets lost at one time and much congestion.

The BIC-TCP algorithm uses a binary search to determine the growth scale of the congestion window. First, it records a maximum point of the congestion window, which is the value of the congestion window when TCP packet loss occurred the last time. A minimum point is also recorded, which is the size of the window when no packet loss events occur in an RTT cycle. A binary search takes the middle point of the minimum and maximum values. When the congestion window grows to this middle value and no packet loss occurs, this indicates that the network can accommodate more packets. Then, this median value is set to the new minimum value and a search for the middle value between the new minimum and maximum values is conducted.

The CUBIC algorithm is the next-generation version of the BIC-TCP algorithm. It greatly simplifies the BIC-TCP window-tuning algorithm by replacing the concave and convex parts of the BIC-TCP window growth with a cubic function, which retains the advantages of BIC-TCP (especially its stability and scalability) while simplifying the window control and enhancing its TCP friendliness.

The CUBIC algorithm’s window growth function can be expressed as
(1)$$W\left(t\right)=C{\left(t-K\right)}^{3}+W_{max}$$
where C is a parameter of CUBIC, t is the time elapsed since the window was last lowered and is an elastic value, and K is the time that the above function increases the current congestion window W to the W_max_ elapsed without further packet loss. The K calculation formula is as follows:(2)$$K=\sqrt[{3}]{\frac{W_{max}*\beta }{C}}$$

#### 2.1.2. Delay-Based CC Algorithms

Delay-based CC algorithms perform link congestion control by measuring the change in delay. The Vegas algorithm takes the increase in RTT as a signal of network congestion. As RTT increases, the congestion window decreases; as RTT decreases, the congestion window increases. The FAST TCP [16] algorithm is a delay-based TCP congestion control algorithm that mainly measures the queue delay in the network as a reference for the degree of network congestion. The delay-based CC algorithm can more accurately judge whether the network is congested than the method based on packet loss. This is because when the data packet loss occurs, the queue has already experienced serious congestion. If CC can be implemented when the router queue starts to jam, it is more conducive to reducing the data packet loss and improving the bandwidth utilization. When there is considerable noise packet loss on the link, the loss-based CC algorithm cannot distinguish between noise packet loss and packet loss caused by router buffer overflow. In this case, the loss-based CC algorithm frequently enters the congestion control phase, resulting in low bandwidth utilization.

#### 2.1.3. Hybrid CC Algorithms

Hybrid congestion control algorithms refer to algorithms based on packet loss and delay. The most representative hybrid CC algorithms are the compound algorithm and BBR algorithm. Compound TCP [17] is a TCP congestion control algorithm implemented by Microsoft. By maintaining two congestion windows at the same time, compound TCP can achieve better performance in a long fat network without loss of fairness. CTCP maintains two congestion windows: a regular additive increase/multiplicative decrease (AIMD) window and a delay-based window, and the actual sliding window size used is the sum of these two windows.

BBR is a congestion control algorithm proposed by Google in 2016 and is based on congestion. The BBR performs congestion control by measuring the bottleneck bandwidth (BtlBw) and round-trip transmission delay (RTprop), which are constantly changing over the life of a connection and must be continually estimated.

At present, the most widely used end-to-end mode CC algorithms are CUBIC and BBR, of which the CUBIC algorithm is the default built-in algorithm of the current mainstream Linux system. The main problem of the CUBIC algorithm is congestion control based on packet loss signals. Once packet loss occurs in the link, the CUBIC algorithm will inevitably reduce the data transmission speed. In addition, packet loss usually occurs after router queue congestion, and the CUBIC algorithm often causes an increase in router queue delay. The BBR algorithm is currently widely used in YouTube’s and Google’s data center servers. The main problems of the BBR algorithm are unfairness in competition with other CC algorithm streams and unfairness in competition with BBR streams (large RTT streams occupy more bandwidth), as well as a slow bandwidth convergence speed [18,19,20].

### 2.2. CC Algorithms Based on the In-Network Control Mode

CC algorithms based on the in-network control mode are also called explicit CC algorithms, and the router informs the data sender of the congestion status in the network. They mainly include the ECN protocol, the XCP protocol, the RCP protocol, and so on.

The explicit congestion notification (ECN) protocol is a standardized explicit control technology. The ECN mechanism informs the sender in advance whether the current link is congested, thereby avoiding packet loss caused by router buffer overflow. Its main disadvantage is that the ECN protocol can only inform the sender whether the link is congested but cannot tell the congestion degree of the link. As a result, the sender cannot accurately control the transmission speed, resulting in insufficient link bandwidth utilization [21].

The explicit control protocol (XCP) is a rate control protocol designed by the Massachusetts Institute of Technology (MIT) in which the router directly feeds the window value to the data sender. The XCP protocol informs the data sender to reduce or increase the window by calculating the load of the link. The XCP protocol adds a 20-byte packet between the IP header and the TCP header, which means that all routers and receivers on the network must support the XCP protocol, which is obviously impossible. In addition, the control of the XCP protocol depends on the estimation of the number of flows, but the number of flows is constantly changing, and its estimation is inaccurate, which will cause network jitter and low link bandwidth utilization [22,23,24].

The RCP (rate control protocol) [25] is an explicit rate control protocol designed by Stanford University. Its core idea is to assign an initial rate to the connection in the TCP handshake stage and then update the rate according to the stream data and feed it back to the data sender. The disadvantage of the RCP protocol is that RCP can only be run on customized RCP routers [26,27] and cannot be run on general routers, so its deployment is difficult, which is also the reason why RCP is not widely used in the network. Similar to the XCP protocol, the number of current data streams cannot be accurately estimated, resulting in a decrease in bandwidth utilization.

### 2.3. SDN and Congestion Control

The key idea of SDN is the separation of management control and data transmission. The logically centralized control plane can control the behavior at the data level by a programming method, which is considered an effective method to manage the complex network environment. Data transmission equipment only needs to follow the controller’s forwarding rules and focus on efficient data forwarding and processing, thus reducing the manufacturing cost and difficulty of data equipment. SDN architecture provides an open programming interface, and managers can realize various functions through programming development. SDN collects network link status and data transmission information through protocols such as OpenFlow(version 1.3) to provide a basis for network management. Administrators use controllers to centrally manage the entire network, send management commands from the application layer to the underlying switch, and direct data transmission at the data level.

The end-to-end mode treats the network as a black box, which cannot solve the fairness problem between flows because flows of different protocols cannot communicate with each other and cannot control each other; for example, BBR protocol flows cannot control the bandwidth of CUBIC flows. The flow between the same protocol can only achieve weak fairness by the design of the algorithm. Once a new flow joins or the network fluctuates, its fairness usually requires multiple RTT times to reach. The end-to-end mode CC algorithm cannot effectively control the queue delay of the link because the data sender cannot obtain the queue situation of the router on the link in real time. The in-network control mode can centrally control the data flow, which can solve the fairness and delay problems that the current end-to-end mode cannot solve. However, in the CC algorithm based on the in-network control mode, only the ECN protocol has been applied in practice, and the ECN protocol can only inform the sender about whether congestion occurs but cannot inform the sender about the degree of congestion and cannot control the transmission speed. The reasons why the CC algorithm based on the in-network control mode has not been applied on a large scale are as follows:Internal network information cannot be accurately obtained.Both XCP and RCP require dedicated routers to apply the protocols, making them difficult to deploy on a large scale.The bandwidth of each stream cannot be accurately controlled (mainly because of inaccurate estimates of the number of flows).

SDN technology can solve the problems existing in the CC algorithm based on the in-network control mode [28,29,30]. We deploy OpenFlow routers inside the network, use OpenFlow controllers to obtain internal information and the network topology, and decouple the network congestion detection module from the congestion control module. The congestion detection module is integrated into the OpenFlow controller, and the congestion control module is integrated into the OpenFlow switch. At present, an increasing number of switches have supported the OpenFlow protocol, which can solve the problem of deploying the CC algorithm based on the in-network control mode. At the same time, we designed an algorithm in the Linux kernel to accurately count the number of flows to allocate the bandwidth of each flow to achieve accurate bandwidth control.

### 2.4. CC Algorithm Evaluation Index

The congestion control algorithm should be evaluated from five aspects: bandwidth utilization, fairness, packet loss rate, convergence speed, and link queue delay. Bandwidth utilization can be expressed as the ratio of the average aggregation speed of a network adapter to the bandwidth value of a network adapter in a period:(3)$$\bar{y_{l}}=\underset{t\to \infty }{\lim}\frac{1}{T}\int_{0}^{T}y_{l\left(t\right)}*dt\approx C$$
where $y_{l\left(t\right)}$ is the aggregation speed (the sum of the throughputs of all data flows) of a network adapter at a certain moment, $\bar{y_{l}}$ is the average speed, C is the bandwidth of the network adapter, and $\frac{\bar{y_{l}}}{C}$ is the bandwidth utilization.

We use the fairness index to evaluate the fairness of the CC algorithm:(4)$$y\left(x_{1},x_{2},\ldots {,x}_{n}\right)=\frac{{\left(\sum_{i=1}^{n}x_{i}\right)}^{2}}{n*\sum_{i=1}^{n}{x_{i}}^{2}}$$
where $x_{i}$ is the bandwidth share of a stream, and the final calculation result ranges from 1/n to 1, with 1 representing the best fairness and 1/n representing the worst fairness.

Packet loss rate is an important indicator to test the quality of congestion control because packet loss not only means that part of the bandwidth is wasted but it will also cause the HOL (head of line) problem, and subsequent TCP packets must be cached to wait for the retransmission of lost packets. This adds additional latency. Routers usually implement a packet loss policy when the buffer overflows. We calculate the packet loss rate by dividing the number of lost packets over a period by the throughput on the router.

The convergence rate of the CC algorithm refers to the time required for the system to reach the target state from the initial state, and the less time required for convergence, the faster the convergence rate. Convergence speed includes the convergence to the target link utilization and convergence to the fairness state. Convergence to the target link utilization refers to the time required for the congestion control mechanism to occupy the idle bandwidth when the link has idle bandwidth. Convergence to fairness refers to the time required for the newly added flow to obtain a fair throughput rate with other flows when the link utilization is close to 100%.

The essential reason for the increase in queue delay is that many packets are piled up in the network cache, resulting in a buffer queue of a certain length. The queue delay can be calculated by measuring the RTT when the network is congested and subtracting the RTT when the network is idle, which is one of the important indicators to evaluate the quality of the congestion control algorithm.

## 3. System Design

In this paper, we propose a congestion control algorithm called BRF based on SDN and utilizing the in-network control mode. The algorithm employs routing feedback to allocate bandwidth for each flow on the router and provide feedback to the data sender. The OpenFlow controller is utilized for gathering network adapter information [13], identifying the bottleneck bandwidth of the network, and deploying OpenSwitch switches at the bottleneck. We deploy OpenSwitch switches in bridge mode, which preserves the existing network topology. The switch runs the Linux kernel with the BRF congestion control algorithm. BRF utilizes two additional bytes in the ACK option field to record feedback bandwidth information for each flow at the bottleneck. This information stores the bandwidth allocated by the bottleneck router and is forwarded after recalculating the checksum. Upon receiving this ACK, the data sender reads and calculates the pacing rate to regulate the data sending rate based on feedback information.

BRF employs a centralized control mode to uniformly allocate flow bandwidth at the bottleneck, thereby ensuring congestion-free network operation while fully utilizing link bandwidth. Furthermore, we propose a hierarchical bandwidth allocation scheme based on an in-network control mode to equitably distribute bandwidth among each flow and categorize all flows according to their maximum data transmission speed. The hierarchy of each flow is dynamically adjusted to prioritize meeting the bandwidth requirements of low-bandwidth flows. We calculate the average bandwidth of a stream based on the quintuple of network connections (source IP, destination IP, source port, destination port, application type), take this bandwidth as the initial bandwidth value of this type of stream, and then allocate an appropriate bandwidth according to the measurement results so that it has a faster convergence speed. By uniformly distributing bandwidth along bottleneck routes, fairness among different streams is ensured. Simultaneously, by leveraging SDN to acquire network card bandwidth information, the unified allocation of bandwidth at the bottleneck route can effectively mitigate queuing delays when the available bandwidth is fully utilized. The BRF protocol can be directly deployed on the bottleneck route or at its entrance/exit using bridge mode. Figure 1 illustrates the deployment locations of the BRF protocol.

### 3.1. SDN-Based Congestion Detection

We use the open source OpenDaylight controller as the controller of the OpenFlow protocol. The southbound interface of the OpenFlow protocol is connected to the OpenFlow switch, and the controller can obtain real-time network status, such as latency, packet loss rate, and available bandwidth. In the OpenDaylight controller, we integrated a congestion detection module to detect whether the link is congested. The architecture diagram of the SDN-based congestion detection module is shown in Figure 2.

The congestion detection module acquires the bandwidth (BW), number of queued packets (P), and length of each queue (Qi, 0 < i < n) to determine if congestion is present by calculating the network adapter utilization and queuing rate. The network adapter utilization can be expressed as follows:(5)$${NetcardRate}_{t}=\frac{(S^{t}-S^{t-∆t})*MSS*8}{BW*∆t}$$

The variable S represents the count of transmitted packets, MSS denotes the maximum length of a packet, and ∆t signifies the duration of time elapsed.

The queue rate can be expressed as
(6)$${Qutilization}_{t}=\frac{P_{t}}{\sum_{i=0}^{n}Q_{i}}$$

When either ${NetcardRate}_{t}\;$> α or ${Qutilization}_{t}\;$> β, the queue is deemed congested. Typically, the values of α and β are 0.98 and 0.1, respectively. The congestion detection module is a very important module in the congestion control algorithm. Even if the most reasonable CC algorithm and bandwidth allocation scheme are set in the network, it is difficult to avoid the emergence of congestion because the network state is constantly changing, and there are constantly new flows joining the network. It is difficult to avoid short-term bursts of large traffic. The congestion detection module is responsible for discovering the network route congestion quickly, reducing the congestion degree, ensuring fairness between flows, and improving the service quality.

The congestion detection module obtains the network status in real time through the network service module in the OpenDaylight controller. When a router is found to be congested, the congestion signal will be sent to the corresponding router to make the router enter the congestion reduction state. The corresponding congestion reduction algorithm will be introduced in detail in Section 3.3.

### 3.2. Protocol Design

We must cache information for all data streams that have not yet terminated their connection on the router and assign an initial bandwidth to each stream. We need to calculate the actual bandwidth occupied by each stream on the router, allocate appropriate bandwidth to each stream through a hierarchical allocation scheme, and write this value into the corresponding acknowledgment response packet before feeding it back to the data sender. The bandwidth occupied by all streams is equal to the total bandwidth value of the network card to ensure that the link does not have congestion and packet loss. At the data sending end, we add a new module in the Linux kernel -TCP_BRF.C module. The main function of this module is to read the bandwidth allocation information in ACK packets and set the data sending speed at the sender end according to the allocated bandwidth value. The setting for the congestion window value (CWND) has been eliminated because it is primarily used for slow start and congestion control in end-to-end protocols. In our proposed BRF mechanism, we have implemented a feedback display and bandwidth allocation system that efficiently handles incoming streams within a short time frame (typically 1–2 RTT). Consequently, the congestion window value is no longer necessary. To illustrate this, let us consider a simple scenario involving a server, a client, an OpenFlow switch, and a controller. The algorithmic flow of the BRF protocol is illustrated in Figure 3.

The protocol process is described as follows:

The hierarchical token bucket (HTB) module is initialized and loaded into the kernel of the switch. The initialization task primarily involves establishing queues (BRF flow queue, CUBIC flow queue, UDP flow queue), configuring flow tables, collecting flow information statistics, and calculating the actual speed of each individual flow. Considering that CUBIC is the prevailing congestion control algorithm of the internet, special processing for CUBIC flows is necessary to ensure fairness among all flows. For a UDP queue flow, a small bandwidth can be allocated and its traffic shaping policy can be set to policing, which involves discarding packets in the event of queuing. The initialization process for the HTB kernel module requires the creation of NIC (network interface card) information structures, queue information structures, and flow information structures, as depicted in Figure 4.
Figure 4NIC queue-stream data structure in the HTB module.
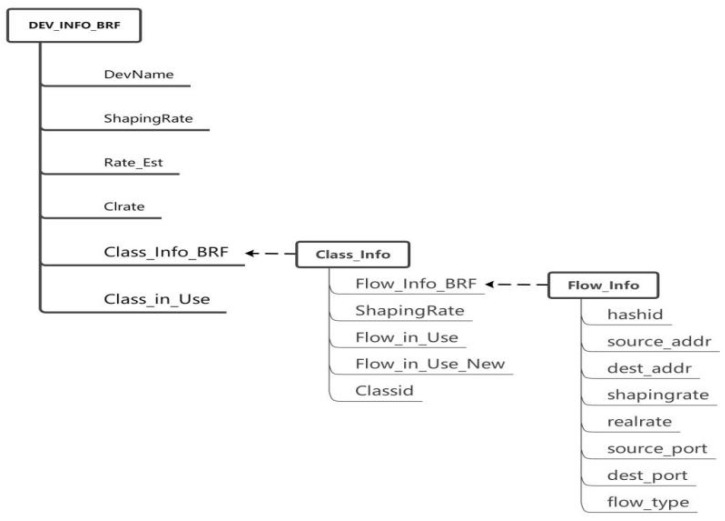
DEV_INFO_BRF stores the information of each NIC. Since there may be multiple NICs in a router, there may be multiple pieces of DEV_INFO_BRF information. CLASS_INFO_BRF is the queue information of an NIC, which mainly includes three types of queues: BRF flow queues, CUBIC and other protocol flow queues, and UDP flow queues. FLOW_INFO_BRF contains specific information for each flow, with the corresponding flow table being inserted upon forwarding of a syn handshake packet by the router and deleted upon connection closure.After the OpenFlow controller obtains the network adapter information (such as bandwidth and queue length) of the switch, the congestion detection module starts to detect the congestion on the network.The sender initiates a TCP handshake. If the stream is identified as a CUBIC stream, it is added to the CUBIC stream queue and modifies the overall queue’s bandwidth allocation value accordingly. In the case of a BRF stream, the initial bandwidth calculation relies on a machine learning model that considers the data stream quintuple (source address, destination address, source port, destination port, protocol type).After receiving the SYN packet, the receiver responds with a reply packet, thus establishing a TCP connection through the three-way handshake protocol.The receiving end replies with an ACK packet, which is the second of three TCP handshakes. The router that forwards the ack packet writes the initial bandwidth of the flow into the ACK packet. The ACK option has two bytes to fill the length. We can use these two bytes to write the bandwidth allocation information. The bandwidth is expressed in bytes and can range from 0 to 65,535. For example, if the allocated bandwidth is 120 mbytes/s = 12,000,000 bytes/s, then two bytes can be represented as 12,005, and 5 means 1200 followed by 5 zeros. The maximum data range we can represent is 0–6,552,000,000,000 bytes/s (0–6552 gbytes/s). The ACK packet of a TCP stream may return to the sending end through the original path or through different paths. Approximately 90% of the streams return through the original path, and each router needs a built-in timer. If no acknowledgment is received for an extended period after adding a new stream to the flow table, it suggests that the feedback acknowledgment may be returned to the sender via a different path. In such cases, a timer is triggered, and the router or switch broadcasts to notify other routers or switches of the insertion operation in the flow table.After receiving the acknowledgment packet, the switch encodes and writes the initial bandwidth allocation information corresponding to the stream into the option field of the acknowledgment packet. Typically, there are two free bytes in the option field of an acknowledgment packet that are utilized for this purpose.After recalculating the checksum of the acknowledgment packet, it is forwarded to the sender. The sender extracts and interprets the bandwidth allocation information from the option field of the ACK packet, subsequently adjusting its data transmission rate accordingly.When subsequent packets of the stream arrive, the actual forwarding speed of the packet is calculated on the switch, and the bandwidth is calculated according to the bandwidth allocation scheme.Upon the arrival of data packets at the receiving end, the newly added stream initiates the transmission of data. The switch then gathers statistics on bandwidth utilization for each stream within every RTT time interval.The receiver continuously transmits ACK packets, and to minimize computing costs, bandwidth allocation information is not included in each ACK packet as it passes through the switch. Instead, this information is added to the ACK packet after every round-trip time (RTT).On the switch, an appropriate bandwidth value is allocated to the stream based on the hierarchical bandwidth allocation scheme, and the value is written into ACK. Section 3.3 describes the specific bandwidth allocation scheme.After the bandwidth is allocated by the bandwidth allocation scheme, the checksum of the ack is recalculated and returned to the data sender.When the OpenFlow controller detects congestion, it informs the switch of the congestion reduction phase.

The state transition diagram for the BRF algorithm is depicted in Figure 5.

In the OpenDaylight controller, we integrated a congestion detection module that performs congestion detection on all OpenFlow switches. The switch first initializes by establishing a flow table. When the link becomes congested, the switch enters a congestion reduction state and uniformly calculates the bandwidth of all streams while reducing queued packets to minimize queuing delays. When the link is in the noncongested state, the initial bandwidth of the newly added traffic is calculated, and then the bandwidth is measured. Then, the bandwidth allocation state is detected. If new traffic is added, the initial bandwidth allocation is performed. After one RTT, the bandwidth allocation phase is entered, and the process is repeated.

The above presents the protocol content of the BRF protocol in a single path, while in multipath mode, it is crucial to ensure that all flows and packets traverse through the OpenFlow switch. A typical multipath application scenario is illustrated in Figure 6.

In multipath mode, the algorithm ensures that all packets and ACKs can pass through the OpenFlow switch. In the returned ACK, it checks whether the free bytes in the ACK option field have been assigned. If no value is assigned, the bandwidth allocation information is written to the ACK directly. If the value is less than or equal to the value carried in ACK, the switch replaces the value carried in ACK. If the value is greater than the value carried in ACK, the switch directly forwards it without modifying the ACK. Assume that there are n routers on the link, and each router allocates BWn bandwidth to a TCP stream. The link bandwidth BWbottleneck that the sender receives from an ACK is expressed as follows:(7)$$BWbottleneck=min\left({BW}_{1},{BW}_{2},\ldots ,{BW}_{1}\right)$$

### 3.3. Bandwidth Allocation Scheme

Data flow refers to the flow of data from the source end to the destination end in network transmission. According to the application layer’s restrictions on the data transmission speed, data flows can be divided into two categories: application-layer restricted flow (app_limited) and application-layer unrestricted flow (app_no_limited).

Application-layer restricted flow often comes from certain applications, such as video websites; in this case, its video buffer is often restricted by the application layer. When a user watches a video, if the network bandwidth is small or there is not enough data in the buffer for other reasons, problems such as stalling and stopping playback may occur. In this case, the user must adjust the cache size or increase the bandwidth and other ways to solve the problem.

In contrast, unrestricted streams at the application layer can be transmitted at maximum speed without any limitations on their velocity. Such data flow is commonly observed in scenarios that demand high real-time and low latency requirements, such as online gaming, voice chat, and other applications that necessitate rapid response times while ensuring stability.

We set the total bandwidth of the NIC of a link to C, and the switch is divided into three queues (BRF flow, CUBIC flow, UDP flow), where the bandwidth of the BRF flow queue is $C_{BRF}$, the bandwidth of the CUBIC flow is $C_{CUBIC}$, the number of BRF flows is $N_{BRF}$, and the number of CUBIC flows is $N_{CUBIC}$; then
(8)$$C_{BRF}=\frac{N_{BRF}}{N_{CUBIC}+N_{BRF}}*C,\;C_{CUBIC}=\frac{N_{CUBIC}}{N_{CUBIC}+N_{BRF}}*C$$

Due to protocol limitations, the bandwidth of each CUBIC flow cannot be accurately controlled. However, a queue can be set for CUBIC at the switch level, and its total bandwidth can be set. The total bandwidth of the queue can be allocated proportionally according to the number of flows to achieve fairness between CUBIC flows and BRF flows.

For BRF flows, the specific bandwidth allocation scheme is as follows:

For the newly added flow i ($i\in N_{BRF}$), the initial bandwidth is allocated according to the network quintuple:(9)$${BW}_{i-init}=F(s_{addr},d_{addr},s_{port},d_{port},type)$$

To ensure fairness, the ${BW}_{i-init}$ value should not exceed the average bandwidth value, which can be expressed a:(10)$${BW}_{i-init}=\left\{\begin{array}{c} {BW}_{init},{BW}_{init}<{BW}_{avg}\\ {BW}_{avg},{BW}_{init}\geq {BW}_{avg} \end{array},{BW}_{avg}=\frac{C_{BRF}}{N_{BRF}}\right.$$

The residual bandwidth RBW is defined as the difference between the total bandwidth of the BRF flow and its utilized bandwidth, which can be mathematically expressed as follows.
(11)$$RBW=C_{BRF}-\sum_{i=0}^{N_{BRF}}{BW}_{i-real}$$

If $RBW\geq {BW}_{i-init}$, then the bandwidth value ${BW}_{i}$ allocated by the newly added flow i and the bandwidth value allocated by other BRF flows can be expressed as follows.
(12)$$\left\{\begin{array}{c} {BW}_{i}={BW}_{i-init}\\ {BW}_{j}={BW}_{j},\left(j\in N_{BRF},j\neq i\right) \end{array}\right.$$

If $RBW<{BW}_{i-init}$, then the bandwidth value ${BW}_{i}$ allocated by the newly added flow i and the bandwidth value allocated by other BRF flows can be expressed as follows.
(13)$$\left\{\begin{array}{c} {BW}_{i}={BW}_{i-init}\\ {BW}_{j}={BW}_{j}*\frac{C_{BRF}-{BW}_{i}}{\sum_{i=0}^{N_{BRF}}{BW}_{i-real}},\left(j\in N_{BRF},j\neq i\right) \end{array}\right.$$

At present, the bandwidth allocation for the new flow has been established, and after a designated period (max(${RTT}_{i}$), $i\in N_{BRF}$), the actual flow rate is measured. If ${BW}_{i-real}<{BW}_{i}*\alpha$, the flow is considered app_limited, while if ${BW}_{i-real}>{BW}_{i}*\alpha$, it is deemed app_no_limited; typically, α falls between 0.95 and 1.0. In this scenario, the bandwidth allocation scheme for flow i is as follows:(14)$${BW}_{i}=\left\{\begin{array}{c} {BW}_{i-real}*\beta ,\beta \in [1.05-1.10],i\in N_{BRF-app-limited}\\ \frac{C_{BRF}-\sum_{i\in N_{BRF-app-limited}}{BW}_{i}}{N_{BRF}-N_{BRF-app-limited}},i\in N_{BRF-app-no-limited} \end{array}\right.$$

At this juncture, the allocation of bandwidth has been finalized, and the phase of measuring bandwidth commences. Once the measurement is completed, it will proceed to the bandwidth allocation phase if no new flows are added or transition to the bandwidth initialization phase if new flows are added, and so on. In the event of link congestion, the system enters a phase of congestion elimination whereby bandwidth allocation is determined based on the type of congestion present. The variable θ represents the specific type of congestion, while $\theta_{1}$ and $\theta_{2}$ represent queuing and instantaneous burst traffic-induced congestion, respectively. The process for reducing such congestion is as follows:

Let t be the time required to eliminate congestion and ∆t be the time interval between the next bandwidth allocation and the previous allocation, which can be expressed as follows.
(15)$$∆t=\max({RTT}_{i}),i\in N_{BRF}$$
where q(t) is the queue length and MSS is the maximum packet segment length; then
(16)$$\left(\sum_{i\in N_{BRF}}{BW}_{i}-\sum_{i\in N_{BRF}}{BW}_{i+∆t}\right)*t=q\left(t\right)*MSS,\;\theta \in \theta_{1}\;$$

From Equation (16), it can be concluded that
(17)$${BW}_{i+∆t}={BW}_{i}+\frac{\gamma *\left(q\left(t\right)*MSS\right)-\sum_{i\in N_{BRF}}{BW}_{i}}{\sum_{i\in N_{BRF}}{BW}_{i}},\;\theta \in \theta_{1}$$
where y is the emptying coefficient, and the lower the value of this coefficient, the faster the congestion is relieved, usually one or two max (${RTT}_{i}$) values.

If congestion is caused by excessive transient burst traffic, the allocated bandwidth is scaled down:(18)$${BW}_{i+∆t}=\frac{\sum_{i\in N_{BRF}}{BW}_{i-real}}{\alpha *C_{BRF}}*{BW}_{i},\theta \in \theta_{2}$$
where α is the maximum allowed link utilization, typically between [0.95, 1.0].

## 4. Performance Analysis

In this section, we employ the Mininet network simulation tool to evaluate the performance of the BRF protocol. Initially, we recompile the Linux kernel and incorporate the tcp_brf.c protocol file into the net/ipv4 directory of the kernel. Subsequently, we configure the congestion control algorithm of the kernel to utilize the BRF algorithm. In the net/sched directory of the kernel, modifications are made to the sch_htb.c file to incorporate the bandwidth allocation scheme described in this paper, followed by recompiling the kernel. Subsequently, after installing Mininet, we integrate the congestion detection algorithm outlined in this paper into the OpenDaylight source code and proceed with recompiling and installing. After configuring the test environment, we utilize Mininet to construct diverse network topologies for different testing scenarios to ensure the efficacy of our tests.

We compare the performance of the protocol on five indicators, including throughput, fairness, packet loss rate, bandwidth convergence speed, and queuing delay. The test protocols include CUBIC, BBR, and BRF.

### 4.1. Throughput Test

Figure 7a depicts the network topology of Mininet utilized for throughput testing, while Table 1 presents the network configurations of routers R0 and R1. When S1’s CC algorithm is set to BRF, router R0 executes the BRF protocol’s hierarchical bandwidth allocation algorithm, whereas the OpenDaylight controller operates the BRF protocol’s congestion detection module. When the CC algorithm on S1 is configured as BBR or CUBIC, the hierarchical bandwidth allocation algorithm of the BRF protocol and congestion detection module on the OpenDaylight controller are disabled at router R0. Router R1 emulates the network conditions of a real 5G network by setting bandwidth to 100 Mbps and latency to 20 ms. The routing buffer is equipped with two settings, namely, 1000 packets and 100,000 packets. We evaluate the throughput performance of three protocols under various packet loss rates ranging from 0.1% to 10%.

The procedure for throughput testing is as follows:The buffer size of R1 is configured as 1000 packets, while the packet loss rate remains at 0.1%.The CC algorithm on S1 is configured to use the BRF protocol, and the BRF hierarchical bandwidth allocation algorithm is executed on R0. Data transmission from S1 to C1 is conducted for a duration of 60 s, during which throughput measurements are taken.R0 ceases the execution of the BRF hierarchical bandwidth allocation algorithm, configures the CC algorithm to CUBIC on S1, and transmits data from S1 to C1 for a duration of 60 s, and the throughput will be recorded.The congestion control algorithm is configured as BBR on S1 and data transmission from S1 to C1 is initiated. The experiment duration is set to 60 s, and the throughput is recorded.The packet loss rate of router R1 is adjusted to various values ranging from 0.1% to 10%, and steps 2 through 4 are repeated for each adjustment of the packet loss rate.The buffer size of router R1 is configured to 100,000 packets, and steps 2 to 5 are repeated for each subsequent update of the buffer size.

The results of the throughput test are presented in Figure 8. The throughput of the BRF protocol and the BBR protocol exhibits a gradual decline with increasing packet loss rate, as depicted in Figure 8a, with a routing buffer size of 1000 packets. Comparatively, the BRF protocol slightly outperforms the BBR protocol. Conversely, the CUBIC protocol experiences a significant decrease in throughput as the link packet loss rate increases due to its reliance on the packet loss signal. The window value will be significantly reduced, and the throughput of the CUBIC protocol will experience a sharp decline in the presence of continuous packet loss. Figure 8b illustrates the throughput performance of all three protocols under a routing buffer size of 100,000 packets. The throughput of the BRF protocol exhibits a gradual decline as packet loss rates increase, while that of the BBR protocol experiences a rapid decrease under similar conditions when routing buffer sizes are large. This is attributed to the fact that the BBR protocol’s ability to estimate RTT becomes less accurate with larger buffers, leading to reduced throughput.

### 4.2. Fairness Test

Currently, the prevailing congestion control protocol in network environments is the CUBIC protocol. While the CUBIC protocol itself does not exhibit fairness issues, it may encounter unfairness problems when competing with other protocols. For instance, there can be unfairness between BBR and CUBIC [31,32] when both flows are simultaneously loaded at a bottleneck. In this scenario, fairness is contingent upon the buffer queue size of the router at the bottleneck. If the buffer queue is substantial, CUBIC flows will occupy a greater portion of bandwidth; conversely, if it is small, BBR will dominate most of the bandwidth. Furthermore, there exists a fairness issue among BBR protocol flows, as those with larger RTT are more likely to acquire bandwidth. There is RTT unfairness in BBR; that is, in the same bottleneck link, BBR flows with large RTT can obtain most of the bandwidth, and BBR flows with small RTT can obtain less bandwidth or even be starved to death. This small drawback can be exploited maliciously, which means that streams with small RTT may not achieve throughput improvement by using BBR. The intraprotocol fairness test mainly compares the fairness between BBR flows and BRF flows. The interprotocol fairness test includes the mixed fairness test between BBR flow and CUBIC flow and the mixed fairness test between BRF flow and CUBIC flow.

The network topology for the intraprotocol fairness test is depicted in Figure 7b, while Table 2 presents the network settings of routers R0, R1, and R2. We perform separate fairness tests within the BBR protocol and BRF protocol at varying RTT values (100 ms and 20 ms). The following steps outline the testing procedure:The BBR protocol is utilized for establishing connections and transmitting data from S1 to C1 via R0 and R2, as well as from S2 to C1 through R1 and R2.The kernel module sch_htb.c is loaded at the bottleneck bandwidth of R2, and the bandwidth allocation scheme is executed. A connection is established using the BRF protocol to transmit data from S1 to C1 via R0 and R2, and another connection is established using the BRF protocol to transmit data from S2 to C1 via R1 and R2.

Figure 9a illustrates the results of fairness tests conducted on BBR flows with varying RTTS, indicating that higher delay flows are allocated more bandwidth. In contrast, Figure 9b demonstrates the fairness test outcomes for BRF flows with different RTTS, revealing a more equitable distribution of bandwidth compared to that of the BBR protocol. These findings confirm that the BRF protocol exhibits superior fairness relative to its counterpart.

The network topology diagram for the interprotocol fairness test is presented in Figure 7c, while Table 3 displays the network settings of routers R0 and R1. The tests for intermixing BBR and CUBIC flows, as well as BRF and CUBIC flows, were conducted with queue lengths of 1000 and 100,000, respectively. The testing procedure was carried out according to the following steps:The link initialization is configured and the link is set up according to Table 3, assigning a buffer size of 1000 packets to R1.The first sender (S1) utilizes the BBR protocol for connection establishment and data transmission to C1, while the second sender (S2) employs the CUBIC protocol for connection setup and data transmission to C1.S1 establishes connections with C1 using the BRF protocol, whereas S2 uses the CUBIC protocol for connection establishment and data transmission to C1.R1 is configured with a buffer size of 100,000 packets, and steps 2 and 3 are repeated.

Figure 10a illustrates the fairness test results when mixing BBR flow with CUBIC flow using a routing buffer size of 1000 packets, while Figure 10b presents these results with a routing buffer size of 100,000 packets. Larger routing buffers result in increased bandwidth allocation for CUBIC flows, whereas smaller buffers allocate most of the bandwidth to BBR flows.

Figure 10c,d depicts fairness test outcomes when combining BRF flow with CUBIC flow using buffer sizes of 1000 packets and 100,000 packets, respectively. These figures demonstrate that when mixed with the CUBIC protocol, BRF exhibits higher fairness compared to BBR.

### 4.3. Packet Loss Rate Test

Figure 7a shows the network topology of the packet loss rate test, and Table 3 shows the network settings of the packet loss rate test. We test the packet loss rate of the three protocols under routing buffer sizes of 1000 packets and 100,000 packets and count the number of packet losses at R0. The packet loss rate can be expressed as
(19)$$R_{drop}=\frac{P_{drop}}{P_{all}}$$
where $R_{drop}$ represents the rate of packet loss, $P_{drop}$ denotes the number of packets dropped by the router, $P_{all}$ indicates the total number of packets transmitted through the router, and Table 4 presents the results of packet loss rate testing.

Based on the test results, neither the BRF nor the CUBIC protocols significantly contribute to active packet loss of the router, whereas the BBR protocol may result in a considerable number of data packet losses and retransmissions when the link buffer is limited [33].

BRF uses in-network control mode to uniformly allocate the bandwidth of each flow at the bottleneck route. The sum of the bandwidth of each flow is equal to the bandwidth of the network card, so there is no queuing or packet loss. The CUBIC protocol uses a cubic polynomial growth curve, and the Wmax value stores the window value of the last packet loss. The window of the sender grows extremely slowly when approaching Wmax, so the CUBIC protocol causes almost no packet loss.

### 4.4. Convergence Speed Test

We evaluate the convergence time to achieve fairness, which refers to the duration required for a newly added flow to attain a relatively equitable throughput compared to other flows when link utilization approaches 100%. Figure 7a illustrates the network topology employed in the convergence speed test, while Table 3 presents the corresponding network configurations. The test procedure is as follows:S1’s congestion control (CC) algorithm is set as the BRF protocol. At time zero, S1 establishes a connection with C1 and initiates data transmission. Subsequently, a new BRF flow is introduced every two seconds until no additional flows are added at eight seconds. The total duration of this experiment spans 60 s.S1’s CC algorithm is then switched to the BBR protocol, and step 1 is repeated.Finally, S1’s CC algorithm is changed to the CUBIC protocol, and step 1 is repeated accordingly.

The outcomes of the convergence speed test are depicted in Figure 11, demonstrating that both BRF and CUBIC exhibit faster convergence speeds compared to BBR.

The BRF protocol is built on the basis of an in-network control mode, and BRF has global information, so it can quickly allocate the appropriate bandwidth to the newly added flow. When a new flow joins, its initial bandwidth can be determined after 1 RTT, and the appropriate bandwidth of the flow can be determined after another measurement, so its convergence speed is about 2xRTT.

The window growth function of the CUBIC protocol is described in Equation (1) in Section 2.1.1, and the publication of its window growth time is in Equation (2). Since the window growth time of the CUBIC protocol is the third square root of the maximum window, it has a faster convergence rate.

The BBR protocol spends most of its time in a probe bandwidth state to estimate bandwidth. The formula of the estimated bandwidth is Probe_BW = pacing_gain*pacing_rate, and the pacing_gain values of the BBR protocol are usually 1.25 (increasing bandwidth), 0.75 (decreasing queuing), and 1 (running smoothly). Due to its slow bandwidth growth, it usually takes multiple RTTS to converge.

### 4.5. Queuing Delay Test

The CUBIC protocol, which is based on packet loss, can increase the link delay. This is because the CUBIC protocol gradually ramps up the data transmission speed until there is a packet loss due to buffer overflow. Consequently, when the link buffer contains many packets, the limitation of router forwarding speed results in an increase in queuing delay. The BBR protocol will cause an increase in queuing delay when it is in the PROBE_BW state, while the BRF protocol will not cause an increase in queuing delay. The queuing delay calculation formula can be expressed as follows.
(20)$$D_{queue}={RTT}_{measurement}-{RTT}_{empty}$$

In this study, we evaluate the impact of different congestion control (CC) algorithms on queuing delay in a network. Specifically, we measure the queuing delay ($D_{queue}$) using the test topology shown in Figure 7a, where ${RTT}_{measurement}$ represents the current measured round-trip delay and ${RTT}_{empty}$ is the round-trip delay when the routing queue is empty. The CC algorithms tested include the BRF protocol, the BBR protocol, and the CUBIC protocol. We conduct tests by establishing connections from S1 to C1 and sending data while measuring round-trip delays once per second for 30 s under each algorithm. Figure 12 shows the queuing delay test results. Our results show that the CUBIC protocol causes a significant increase in queuing delay, while BBR’s queuing delay increases periodically. In contrast, the BRF protocol does not add any additional queuing delay.

## 5. Conclusions and Future Work

To enhance the bandwidth utilization of 5G mobile networks across various scenarios, ensure fairness among different protocol flows, and minimize link delay, we propose an SDN-based in-network congestion control algorithm. Our approach employs OpenFlow to gather bottleneck bandwidth information within the network and integrates a congestion detection module into the SDN controller to handle burst traffic and rapidly alleviate congestion. We introduce an SDN-based centralized bandwidth allocation scheme to promote fairness among diverse protocol flows. By centrally controlling each flow at the bottleneck bandwidth and rationally allocating bandwidth, our method reduces queuing delay and packet loss caused by routing buffer overflow while maximizing bandwidth usage and improving flow convergence speed. Simulation results demonstrate that our BRF protocol outperforms commonly used CUBIC and BBR protocols in terms of bandwidth utilization, fairness, convergence speed, packet loss rate, and queuing delay. Specifically, the throughput of the BRF protocol is 25.6% higher than that of the BBR protocol and 912% higher than that of CUBIC flow (buffer size = 100,000 packet, packet loss = 2%, RTT = 20 ms). The fairness index of BRF flow when competing with CUBIC flow is close to 1, while the fairness index of BBR flow when competing with CUBIC protocol flow is 0.625, showing the better fairness of the BRF protocol. The convergence time of BRF flows is 30 percent of that of BBR flows.

In future work, we will explore more complex simulation scenarios to analyze our algorithm’s performance further while testing it in real-world settings. We also need to explore the efficiency and security issues of the BRF protocol in the 6G scenario [34]. In addition, we will improve our design to achieve better bandwidth allocation strategies in the case of multiple different competing CC algorithms.

## Figures and Tables

**Figure 1 sensors-24-00749-f001:**
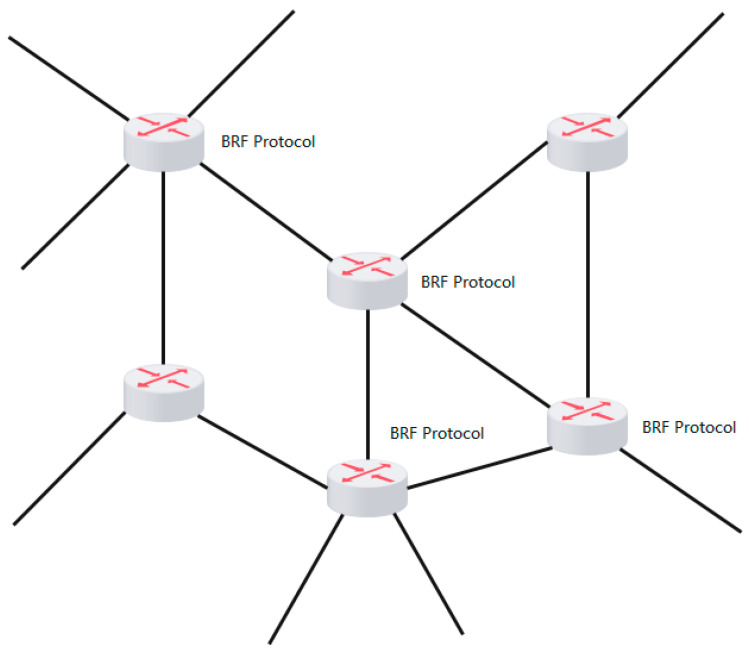
Deployment site of the BRF protocol.

**Figure 2 sensors-24-00749-f002:**
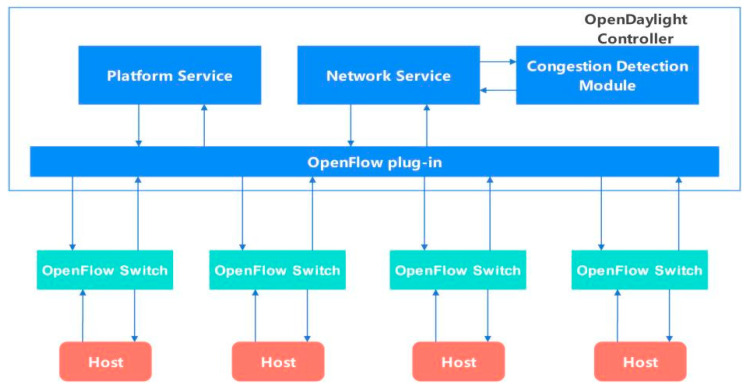
Architecture diagram of the SDN-based congestion detection module.

**Figure 3 sensors-24-00749-f003:**
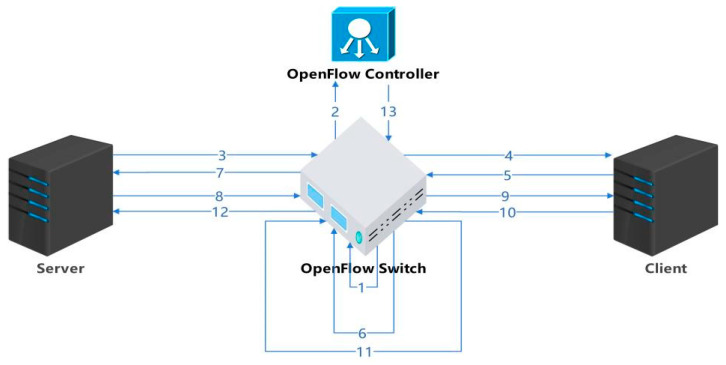
BRF protocol flow.

**Figure 5 sensors-24-00749-f005:**
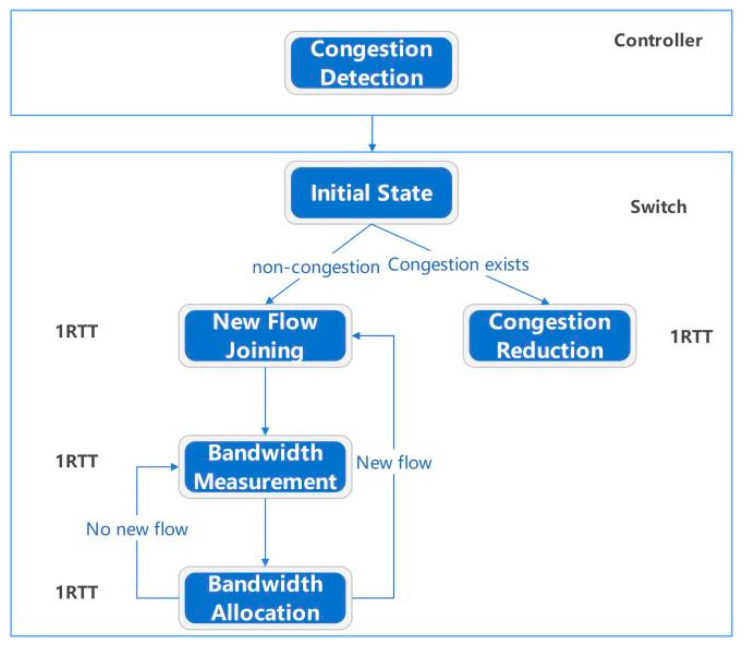
State transition diagram of the BRF algorithm.

**Figure 6 sensors-24-00749-f006:**
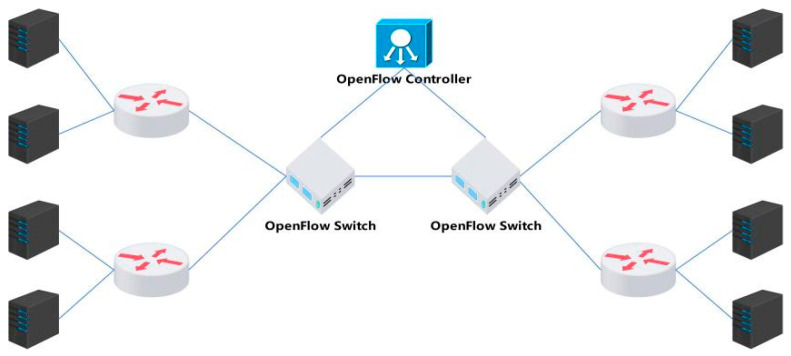
Multipath mode.

**Figure 7 sensors-24-00749-f007:**
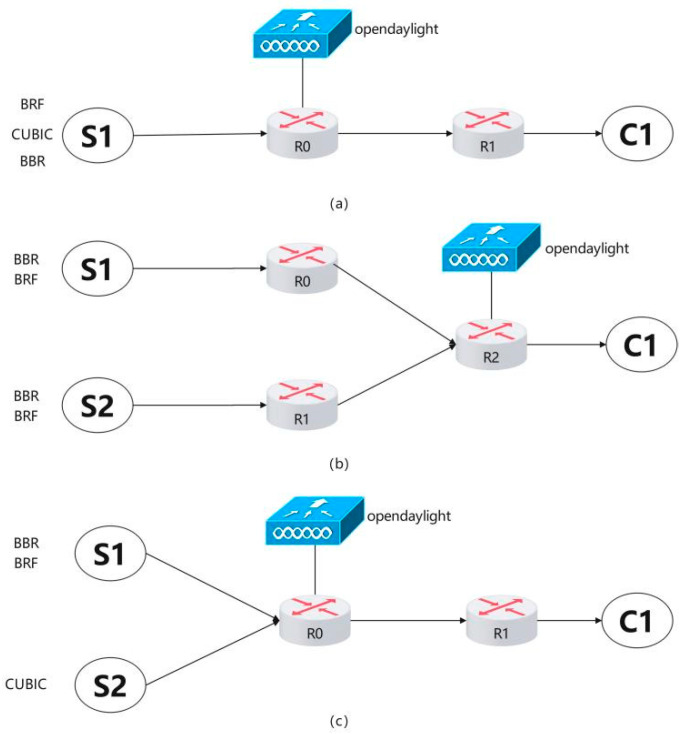
Network topology for protocol testing.

**Figure 8 sensors-24-00749-f008:**
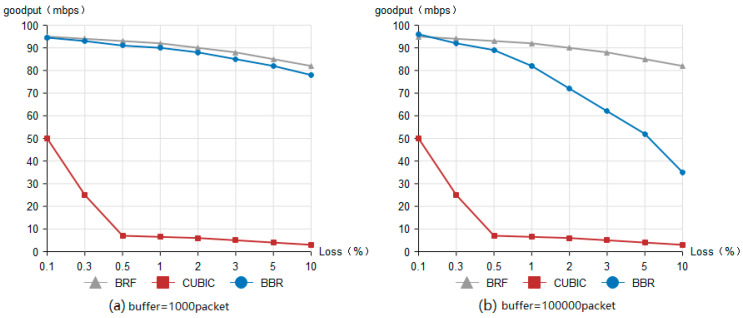
Throughput test result.

**Figure 9 sensors-24-00749-f009:**
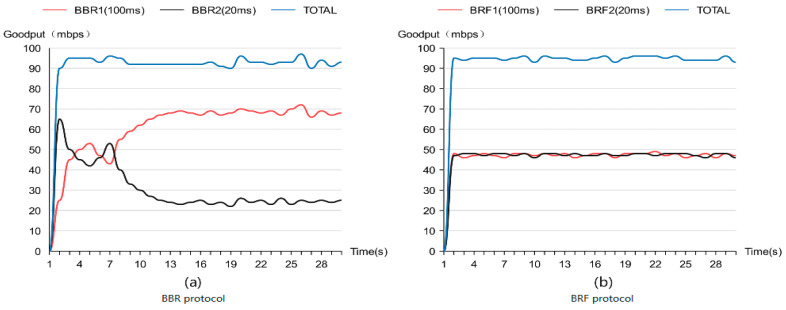
Intraprotocol fairness test results.

**Figure 10 sensors-24-00749-f010:**
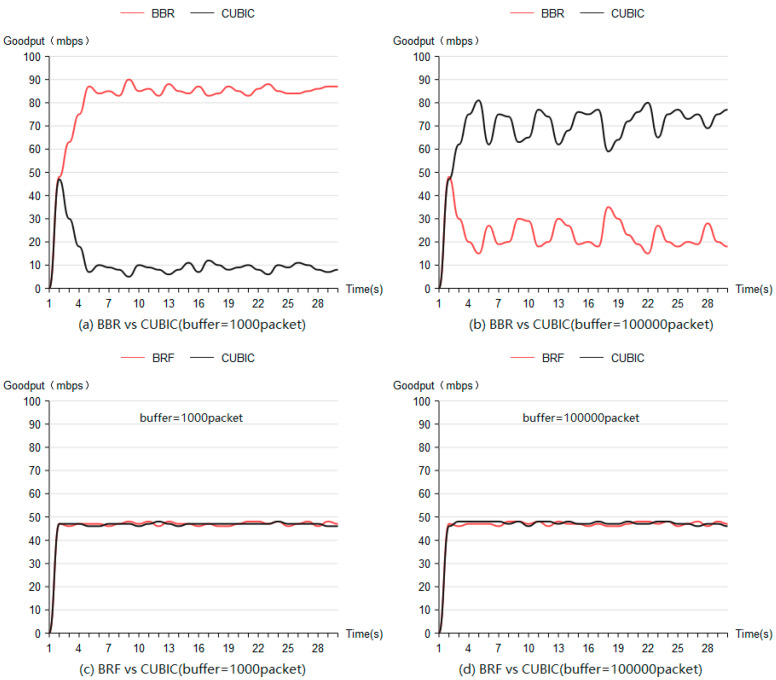
Interprotocol fairness test results.

**Figure 11 sensors-24-00749-f011:**
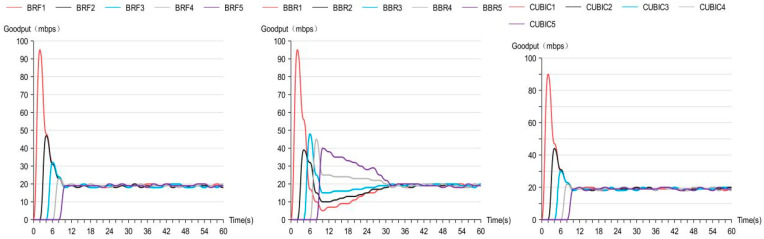
Convergence speed test results.

**Figure 12 sensors-24-00749-f012:**
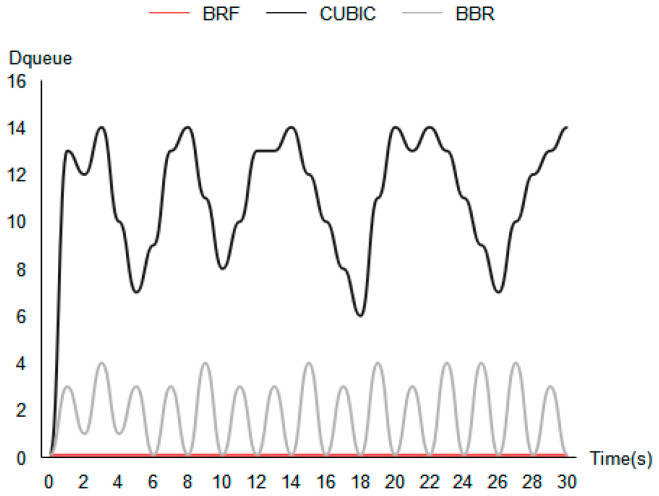
Queuing delay test results.

**Table 1 sensors-24-00749-t001:** Throughput test network setup.

	R0	R1
Capacity	100 Mbps	100 Mbps
Delay	0 ms	20 ms
Loss	0%	0.1–10%
Buffer Size	100,000 packet	1000–100,000 packet

**Table 2 sensors-24-00749-t002:** Intraprotocol fairness test network setup.

	R0	R1	R2
Capacity	100 Mbps	100 Mbps	100 Mbps
Delay	100 ms	20 ms	0 ms
Loss	0%	0%	0%
Buffer Size	10,000 packet	10,000 packet	10,000 packet

**Table 3 sensors-24-00749-t003:** Interprotocol fairness test network setup.

	R0	R1
Capacity	100 Mbps	100 Mbps
Delay	0 ms	20 ms
Loss	0%	0%
Buffer Size	100,000 packet	1000–100,000 packet

**Table 4 sensors-24-00749-t004:** Test results of packet loss rate.

	Buffer = 1000 Packet	Buffer = 100,000 Packet
BRF	≈0%	≈0%
CUBIC	≈0%	≈0%
BBR	≈1.1%	≈0%

## Data Availability

The authors are unable or have chosen not to specify which data have been used.

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
