# Peer review of "SDN-Based Congestion Control and Bandwidth Allocation Scheme in 5G Networks"

_sensors, 2024, doi:10.3390/s24030749_

Round 1

Reviewer 1 Report

Comments and Suggestions for Authors

1) Compare your results with the results of the existing literature.

2)  Why BRF and CUBIC exhibit faster convergence speeds compared to BBR ? Justify.

3) It has been mentioned in Future Work that more complex simulation scenarios will be explored. What types of complex scenarios authors are referring to ? 

4)  In CC Algorithm Evaluation Index, the modeling has been framed by the authors or outsourced from the existing literature. If outsourced, cite the sources.  

5) What is the significance of the fairness index ?

6) Read and use the following relevant reference in your research paper :

Anand, R., Ahamad, S., Veeraiah, V., Janardan, S. K., Dhabliya, D., Sindhwani, N., & Gupta, A. (2023). Optimizing 6G Wireless Network Security for Effective Communication. In Innovative Smart Materials Used in Wireless Communication Technology (pp. 1-20). IGI Global.

Reviewer 2 Report

Comments and Suggestions for Authors In this paper, autho proposes a congestion control algorithm BRF using an in-network control mode based on bottleneck routing feedback and SDN. It clear research done on the SDN-Based Congestion Control and Bandwidth Allocation Scheme in 5G Networks congestion control algorithm combining SDN and an in-network control mode and a hierarchical bandwidth allocation scheme. The first part aims to improve bandwidth utilization and reduce queueing delays. The second part aims to ensure the fairness of each flow and improve the convergence speed

Following topic address specific gap to improve the bandwidth utilization of 5G networks。 To ensure the fairness of each flow and the speed of flow convergence, Author conducts experiments with Mininet. Author compared BRF with existing CC algo- 118 rithms, including CUBIC and BBR, in terms of bandwidth utilization, queue delay,,,

* BRF based on SDN  already exciting what is the novelty in the the work?

* How BWbottleneck measure?

* In Bandwidth Allocation Scheme what is used for this work?

* Table 1. format need to check.

* If possible author consolidated Performance Analysis in single table.

* Conclusion please add the numerical result. 

Reviewer 3 Report

Comments and Suggestions for Authors

The paper proposed a congestion control algorithm named BRF. It uses routing feedback from bottlenecks to enhance the bandwidth allocation mechanism and provide a better convergence rate. The contribution of the article is clear. However, there are some points for authors to address:

The last paragraph in section 1 should reflect the actual used numbering.

It would be better if you added a diagram illustrating the modified format of the ack msg and what information are passed through these two additional bytes in the ack option field.

Some of the data flows in Figure 1 are not clear. Please use darker colors or borderline to depict the flows.

A pseudocode for the proposed bandwidth allocation scheme would make the algorithm clearer for readers.

Regarding the packet loss rate result, analyze the result and explain why the BRF and CUBIC do not cause traffic loss.

 Similarly, more explanation is required in the convergence speed result. Why did the BBR fail? And why the proposed system did not not give a better result compared to CUBIC?

Correct the legend in Figure 12. It has a duplication of the BRF title.

The first time you use an abbreviation in the text, present both the spelled-out version and the short form. For example: rtt.

Please check for all other abbreviations in your manuscript.

Include more recent references. 

Reviewer 4 Report

Comments and Suggestions for Authors

see annex

Comments on the Quality of English Language

For some sentence grammar to be modified
